# Fantastic Experts and How to Find Them: A Multi-Dimensional Study for Experts-Level Sparsification in Mixture-of-Experts

## Abstract

Sparsely activated Mixture-of-Experts (SMoE) has shown promise in scaling up the learning capacity of neural networks. However, vanilla SMoEs have issues such as expert redundancy and heavy memory requirements, making them inefficient and non-scalable, especially for resource-constrained scenarios. Expert-level sparsification of SMoEs involves pruning the least important experts to address these limitations. In this work, we aim to address **three** questions: ① What is the best recipe across multiple plausible recipes to identify the least knowledgeable subset of experts that can be dropped to achieve a desired sparsity level? ② How should we perform expert dropping (one-shot or iterative), and what correction measures can we undertake to minimize its drastic impact on SMoE subnetwork capabilities? ③ What capabilities of full-SMoEs are severely impacted by the removal of the least dominant experts, and how can we recover them? *Firstly,* we propose **MoE Experts Compression Suite (MC-Suite)**, which is a collection of some previously explored and multiple novel recipes to provide a comprehensive benchmark for estimating expert importance from diverse perspectives, as well as unveil numerous valuable insights for SMoE experts. *Secondly,* unlike prior works with a one-shot expert pruning approach, we explore the benefits of iterative pruning with the re-estimation of the MC-Suite criterion. Moreover, we introduce the benefits of task-agnostic fine-tuning as a correction mechanism during iterative expert dropping, which we term **MoE Lottery Subnetworks**. *Lastly,* we present an experimentally validated conjecture that, during expert dropping, SMoEs' instruction-following capabilities are predominantly hurt, which can be restored to a robust level subject to external augmentation of instruction-following capabilities using k-shot examples and supervised fine-tuning.

## 1 Introduction

Sparsely activated Mixture-of-Experts (SMoEs) are a promising architecture design that facilitates an amalgamation of the collective intelligence of multiple experts and are distinguished by their ability to dynamically allocate computational resources based on the input. Mixture-of-Experts, initially introduced in (Shazeer et al., 2017a), has undergone extensive exploration and advancement, and is now adopted in industry-scale LLMs (*e.g.*, Mixtral-8×7B, Grok-1, DBRX, *etc.*), achieving stellar performance across various NLP and CV task leaderboards. Despite the sparse nature of MoEs promising enhanced efficiency and scalability, they have crucial limitations: ① SMoEs trade space for FLOPs, which require high memory usage due to the duplication of the network layers into

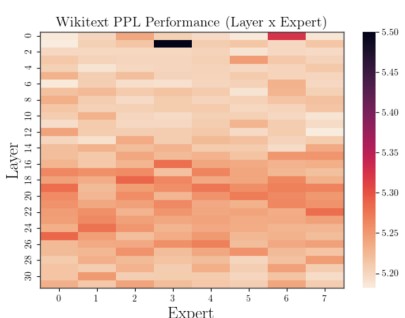

Figure 1: Wikitext Perpelxity of Mixtral 8×7B pretrained checkpoint when removing a single expert $e$ from layer $l$.

multiple copies as experts; ② SMoEs tend to have poor utilization of their capacity and existence of redundancy (Mittal et al., 2022; Chen et al., 2023) due to representation collapse.

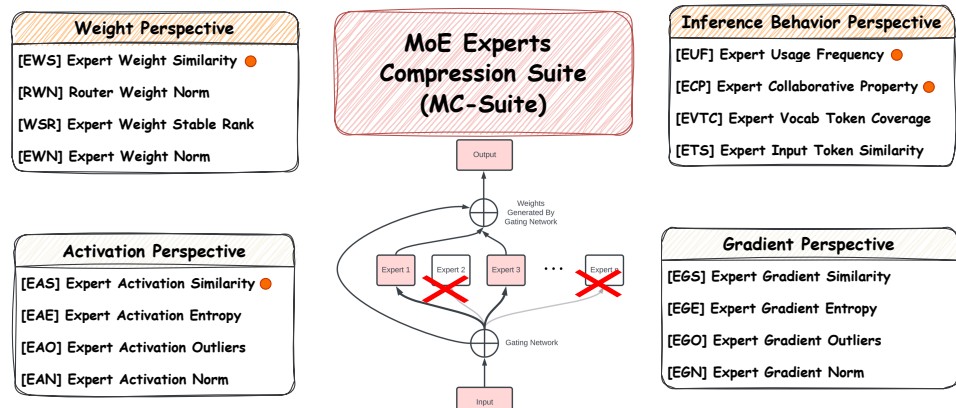

Figure 2: **MoE Experts Compression Suite (MC-Suite):** A comprehensive basket of criterions ($c$) to investigate dominant experts across different SMoE blocks from *weight, expert behavior, intermediate activations, and gradient behavior perspective*. Criterion with 🟠 indicate it has been previously explored either in exactly the same formulation or with slight variation. Based on the score of a criterion ($\text{score}_c^e$) estimated within a MoE layer, an expert ($e$) is identified and removed.

In parallel to well-studied techniques that address memory and compute bottlenecks using weight sparsity (Jaiswal et al., 2023c; Lee et al., 2019; Frankle & Carbin, 2019; Yin et al., 2023b; Liu et al., 2023a) and quantization (Liu et al., 2023b; Kim et al., 2023; Dettmers et al., 2023; Frantar et al., 2022; Lin et al., 2023), SMoEs architecture design facilitates a unique opportunity for *expert-level sparsification* that aims to compact the SMoE model by retaining fewer but more knowledgeable experts. For instance, Figure 1 illustrates that the existence of some experts is *critically important (dominant)* and dropping them could lead to an abrupt performance drop, while some experts are notably redundant with negligible impact when removed. Recently, a few works[1] have proposed expert importance estimation techniques such as token reconstruction loss (Lu et al., 2024) and heavy-hitters counting (Muzio et al., 2024), illustrating the potential of expert dropping. However, a comprehensive benchmarking of possible task-agnostic recipes to select the best recipe is still missing. At this point, one key question arises: *What is the best recipe to identify less knowledgeable experts that can be dropped without sacrificing the vital knowledge and capabilities of the SMoE?*

In this work, we present **MoE Experts Compression Suite (MC-Suite)**, a comprehensive collection of potential recipes for *expert importance estimation*[2] which studies "clues" from four broad and diverse perspectives: ⓐ expert & router weight dynamics, ⓑ expert inference behavior dynamics, ⓒ intermediate activation properties, and ⓓ expert gradient properties. In addition to expert importance, MC-Suite unveils numerous valuable insights across experts: dominant experts tend to have lower stable-rank (*i.e.,* pretraining knowledge is well compressed (Jaiswal et al., 2024)) which is favorable for additional compression using low-rank factorization; intermediate activation and gradients corresponding to dominant experts tend to have higher entropy indicating better information quantity and conducive finetuning abilities for downstream adaptation (Zhang et al., 2024; Zhao et al., 2024); among many others as outlined in Section 3. It is important to note that dropping experts involves deleting its entry in the router gating function, which leaves the MoE subnetwork in a sub-optimal state (*i.e.,* increased skewness in load distribution across retained experts, abrupt drop in performance with high dropping ratio). Most existing prior works (Lu et al., 2024; He et al., 2024; Muzio et al., 2024) adopt *one-shot* criterion estimation for expert removal that alleviates the impact incurred due to sparsification in the form of load imbalance and abrupt performance drop.

In this work, we systematically illustrate that extending one-shot pruning to iterative pruning with re-estimation of importance criterion in $k$-rounds[3], leads to identifying a better subset of experts for dropping. Moreover, motivated by lottery ticket hypothesis (Frankle & Carbin, 2018), we pro-

---

[1]Performance comparison of best recipe from MC-Suite w.r.t. SoTA MoE expert pruning techniques (Lu et al., 2024; Muzio et al., 2024) is present in Appendix A.5 for MMLU task.

[2]MC-Suite is composed of multiple novel proposed criterions (*e.g.,* entropy-based, norm-based) as well as prior explored criterions (*e.g.,* expert usage, expert weight similarity).

[3]Our ablation in Appendix A.3 illustrate that subnetworks idetified from one-shot vs. iterative pruning are significantly different. We conclude that iterative pruning helps in improving subnetwork quality while additional finetuning helps in retaining the capabilities of subnetwork to avoid abrupt performance degradation.

pose **MoE Lottery Subnetwork** which involves *task-agnostic budget finetuning*[4] using next-token prediction objective to address the intermediate sub-optimal state induced due to expert-level sparsification. More specifically, the MoE lottery subnetwork is derived using an iterative *estimation-prune-finetune* procedure, and our experiments illustrate that the task-agnostic finetune submodule can help in load distribution across remaining experts along with improving the performance.

To unveil the true merits of expert-level sparsification, in this work we ask an interesting question: *Given the existence of redundancy across experts, during expert-level sparsification, what capabilities of full-MoE are severely impacted?* We hypothesize that during expert-level sparsification of well-trained MoEs, *instruction following capabilities* are **notably hurt** while the derived MoE subnetwork still retains the pretraining knowledge and reasoning abilities to a great extent. Our work design controlled experiments from zero-shot setting to $k$-shot setting and supervised finetuning (SFT) using instruction-tuning dataset, to augment instruction following capabilities into derived MoE subnetwork. Our experimental results indicate that external instruction-following support can impressively minimize the performance drop due to expert-level sparsification on complex reasoning downstream tasks. Our key contributions can be briefly summarized as:

- We present **M**oE **E**xperts **C**ompression **Suite (MC-Suite)**, to re-look the expert importance estimation and facilitate a comprehensive benchmark from a multi-dimensional perspective. Our extensive experiments show that activation & gradient-guided importance estimation criterions that take into account both input tokens and weight parameters, identifies a superior subset of least dominant experts which can be dropped with minimal impact.

- We explore the potential of iterative **estimate-prune-finetune** procedure in context of expert-level sparsification. Our experiments illustrate that a fairly limited amount of task-agnostic finetuning facilitate not only improved performance of resultant subnetwork but overcome the skewness in load distribution incurred due expert dropping.

- Our extensive experiments across multiple downstream dataset (*e.g.,* MMLU, ARC-c, ARC-e, HellaSwag, and WinoGrande) surprisingly found that MoE subnetworks, even at non-trivial sparsity ratios (*e.g.,* $\geq 50\%$ with $\geq 1.27\times$ speedup and $\leq 0.55\times$ memory usage) can achieve **robust** performance subjected to external augmentation of instruction following capabilities using $k$-shot examples or supervised finetuning.

## 2 MoE Experts Compression Suite (MC-Suite): An Exhaustive Basket of Strategies to Find Fantastic Experts

Mixture-of-Experts (MoE) architecture has been recently gaining enormous attention for the scaling up of LLMs while maintaining roughly constant FLOPs. By incorporating multiple expert networks and employing a sparse gating mechanism, MoE achieves efficient computation, enabling the development of larger models within the constraints of limited computational resources (Fedus et al., 2022; Jiang et al., 2024). Despite its advantages, MoE suffers from extensive memory costs, which hinder its practical deployment and widespread adoption. For example, the Mixtral-8×7B MoE model takes around 180GB memory while only 28GB parameters are activated for each input token[5]. In parallel to conventional model compression techniques like weight sparsity, quantization, and distillation; the architecture design of MoEs facilitates a unique opportunity for *expert-level sparsification* which involves identifying and removing the least important experts or connections.

Figure 1 presents the wikitext perplexity of Mixtral-8×7B by dropping a single expert $e$ from layer $l$. It can be clearly **noted** that some experts tend to have an abrupt impact on the performance of the pre-trained checkpoint compared to others[6]. Therefore, it is critically important to carefully identify the subset of *least important* experts, which are pruned to match the desired sparsity level with minimal impact on performance. In this section, we present **M**oE **E**xperts **C**ompression **Suite (MC-Suite)**, a **first** comprehensive benchmark to investigate expert importance using a wide spectrum of novel and previously explored (*e.g.,* expert usage frequency) criterions broadly categorized in four

---

[4]Our experiments in Appendix A.2 confirms that a limited number of training iterations are sufficient to address the sub-optimal state of MoE subnetwork produced after expert-level sparsification.

[5]The estimates are calculated using full precision (float32).

[6]Some Experts are Special: Across our experiments, we found that dropping of special experts lead to abrupt performance drop and this behaviour is consistent for different tasks and datasets.

groups: weight-guided expert importance, inference behavior based importance, activation-guided importance, and gradient-guided importance.

## 2.1 PRELIMINARIES AND NOTATIONS

Consider an MoE-based transformer model $M_L$ with $L$ MoE layers for processing a set of input tokens $\mathcal{X} = \{x_1, x_2, ..., x_t\}$. A standard MoE layer ($M_l$) is composed of a set of $n$ experts $\mathcal{E} = \{E_1, E_2, ..., E_n\}$ with corresponding weights $\{W_1, W_2, ..., W_n\}$ and a gating function $G$ with weight matrix $W_G^{d \times n}$. The gating function is responsible for selecting which experts will be activated for a given input token $x_i$ by estimating selection score $G(x_i) \in^n$ with respect to all experts in $\mathcal{E}$. The input token $x_i$ is processed by top-$k$ experts with scaled highest score, and the expert's outputs (intermediate activations) $\mathcal{A} = \{a_1, a_2, ..., a_k\}$ are combined into a weighted sum based on affinity score provided by the gating function. It can be summarized as follows:

$$\mathcal{K}_i = \text{top-}k(\text{softmax}(G(x_i)), k) \tag{1}$$

$$y_i = \sum_{m \in \mathcal{K}_i} G_m(x_i) \cdot E_m^{W_m}(x_i) \tag{2}$$

where $\mathcal{K}_i$ indicated the top-$k$ indices of the selected experts for token $x_i$, $G_m(x_i)$ and $E_m^{W_m}$ represents the affinity score and output for $m$-th expert for token $x_i$.

## 2.2 WEIGHT-GUIDED EXPERT IMPORTANCE

① **Expert Weight Similarity Criterion (EWS):** In this criterion, we flatten the weights of all experts of layer $l$ of $M$ and calculate pairwise cosine similarity across them. Depending on the `min` or `max` argument, we select expert $E_p$ which have min or max cosine similarity with $\mathcal{E} - \{E_p\}$.

$$\cos_{n \times n} = \texttt{pairwise-cosine}_{\forall (p,q) \in \mathcal{E} \times \mathcal{E}}(\texttt{flatten}(W_{E_p}))$$
$$\text{drop-index} = \texttt{min/max}_{\forall p \in \mathcal{E}}\big\{\texttt{sum}(\cos_{[p,:]}) - \cos_{[p,p]}\big\} \tag{3}$$

② **Router Weight Norm Criterion (RWN):** Given a token, the router gating function is responsible for selecting top-$k$ experts from $n$ available experts using its weight matrix $W_G^{d \times n}$. RWN aims to understand the role of the gating weights corresponding to $E_p$ in $W_G$ to estimate its importance.

$$\text{drop-index} = \texttt{min/max}\big\{\text{norm}_{l2}(W_G^{d \times n}, \text{dim=1})\big\} \tag{4}$$

③ **Expert Weight Stable Rank Criterion (WSR):** Stable rank of an expert weight matrix ($W_{E_p}$) is defined as $\frac{\sum_{i=1}^{r} \sigma_i^2(W_{E_p})}{\sigma_1^2(W_{E_p})}$, where $\sigma_i$ refers to the $i$-th sorted singular value of $W_{E_p}$. Recently, stable-rank has been studied in the context of LLM layer importance, generalizability, and downstream adaption ability (Sanyal et al., 2020; Jaiswal et al., 2024; Zhang et al., 2024) and we aim to extend it for estimation of expert importance.

$$\text{drop-index} = \texttt{min/max}\big\{\texttt{stable-rank}_{\forall p \in \mathcal{E}}(W_{E_p})\big\} \tag{5}$$

④ **Expert Weight Norm Criterion (EWN):** In this criterion, we calculate the $l2$-norm of weights of all experts of layer $l$ of model $M$. Depending on the `min` or `max` argument, we select expert $E_p$ that has min or max weight norm for dropping.

$$\text{drop-index} = \texttt{min/max}\big\{\text{norm}_{l2}(W_{E_p})\big\} \tag{6}$$

## 2.3 INFERENCE-GUIDED EXPERT IMPORTANCE

① **Expert Usage Frequency Criterion (EUF):** In this criterion, we define expert usage with a calibration dataset (*e.g.*, C4 validation for MC-Suite). Expert usage is estimated by the ratio of tokens that activate $E_p$ with a fixed calibration set. Note that we experimentally found that expert

usage frequency is **not** strongly tied to the choice of calibration dataset. Given $\mathcal{X}$ as calibration set with $t$-tokens and $\mathcal{K}_i$ be the top-$k$ experts for token $i$, we select expert $\boldsymbol{E}_p$ as:

$$\text{drop-index} = \min/\max_{\forall p \in \mathcal{E}} \Big\{ \sum_{x_i \in \mathcal{X}} \mathbb{1}[\mathcal{K}_i \cap \{E_p\}] \neq \varnothing \Big\} \tag{7}$$

② **Expert-Expert Collaboration Criterion (ECC):** Expert-Expert collaboration count is as defined as the number of times two experts $\boldsymbol{E}_p$ and $\boldsymbol{E}_q$ are selected to process a token $x_i$. Let $\mathcal{X}$ as calibration set with $t$-tokens and $\mathcal{K}_i$ be the top-$k$ experts for token $i$, we define:

$$\text{collaboration-matrix}_{(\boldsymbol{E}_p, \boldsymbol{E}_q) \in (\mathcal{E} \times \mathcal{E})}^{n \times n} = \sum_{x_i \in \mathcal{X}} \mathbb{1}[\mathcal{K}_i \cap \{E_p, E_q\} == \{E_p, E_q\}] \tag{8}$$

Given the collaboration matrix, we select the expert pair $(\boldsymbol{E}_p, \boldsymbol{E}_q)$ wrt. the `min` or `max` argument and drop-index is identified as the expert that tends to have lower usage frequency.

③ **Expert Vocabulary Coverage Criterion (EVTC):** Expert vocabulary coverage is defined as the fraction of unique tokens from the model vocabulary, which is processed by a given expert $\mathcal{E}_p$. Consider $\mathcal{V}$ be the model vocabulary and $\mathcal{X}_p$ are the tokens from calibration set $\mathcal{X}$ which are routed to expert $\boldsymbol{E}_p$ by gating function, we select $\boldsymbol{E}_p$ as:

$$\text{drop-index} = \min/\max_{\forall p \in \mathcal{E}} \big\{ \texttt{unique}(\mathcal{X}_p)/|\mathcal{V}| \big\} \tag{9}$$

④ **Expert Input Token Similarity (ETS):** In this criterion, we aim to estimate the input token-level similarity across experts. More specifically, with $\mathcal{X}_p$ as the tokens routed to expert $\boldsymbol{E}_p$, we generate:

$$\text{token-similarity-matrix}_{(\boldsymbol{E}_p, \boldsymbol{E}_q) \in (\mathcal{E} \times \mathcal{E})}^{n \times n} = \texttt{count}(\mathcal{X}_p \cap \mathcal{X}_q) \tag{10}$$

Given the token similarity matrix, we select the expert pair $(\boldsymbol{E}_p, \boldsymbol{E}_q)$ wrt. the `min` or `max` argument and drop-index is identified as the expert that tends to have lower usage frequency.

## 2.4 ACTIVATION-GUIDED EXPERT IMPORTANCE

① **Expert Activation Similarity Criterion (EAS):** Given the calibration set of tokens $\mathcal{X}$, we accumulate the activation of tokens routed to experts ($\mathcal{A}_{\boldsymbol{E}_p}$) using forward hooks. We first generate the activation similarity matrix across each expert pair depending on `min` or `max` argument, we select expert $\boldsymbol{E}_p$ which have min or max activation similarity with $\mathcal{E} - \{\boldsymbol{E}_p\}$.

$$\text{activation-similarity}_{(\boldsymbol{E}_p, \boldsymbol{E}_q)}^{n \times n} = \frac{1}{|\mathcal{A}_{\boldsymbol{E}_p}| \times |\mathcal{A}_{\boldsymbol{E}_q}|} \sum_{(a_m, a_n) \in (\mathcal{A}_{\boldsymbol{E}_p} \times \mathcal{A}_{\boldsymbol{E}_q})} \texttt{cosine}(a_m, a_n) \tag{11}$$

$$\text{drop-index} = \min/\max_{\forall p \in \mathcal{E}} \big\{ \texttt{sum}(\text{activation-similarity}_{[p,:]}) - \text{activation-similarity}_{[p,p]} \big\}$$

② **Expert Activation Entropy Criterion (EAE):** Entropy is the measurement of information quantity and we extended (Lin et al., 2024) entropy quantification strategy for convolution feature maps to expert activation. More specifically, in MC-Suite, the entropy of an expert activation ($\mathcal{A}_{\boldsymbol{E}_p}$) is proportional to the summation of the logarithm of the standard deviation of each hidden dimension:

$$H(\mathcal{A}_{\boldsymbol{E}_p}) \propto \sum_j \texttt{log}[\sigma(\mathcal{A}_{\boldsymbol{E}_p}^j)] \tag{12}$$

where, $\sigma(\mathcal{A}_{\boldsymbol{E}_p}^j)$ calculate the standard deviation of $j_{th}$ hidden dimension of the activation and sum it to obtain activation entropy and select expert $\boldsymbol{E}_p$ which have min or max activation entropy.

③ **Expert Activation Distribution Outliers (EAO):** In this criterion, we estimate outliers in the normally distributed activation of experts. More specifically, given $\mathcal{A}_{\boldsymbol{E}_p}$ as the activations of expert $\boldsymbol{E}_p$, we estimate mean ($\mu_{\mathcal{A}_{\boldsymbol{E}_p}}$) and standard deviation ($\sigma_{\mathcal{A}_{\boldsymbol{E}_p}}$) across the hidden dimension and count outliers outside the interval $\mu_{\mathcal{A}_{\boldsymbol{E}_p}} \pm c \times \sigma_{\mathcal{A}_{\boldsymbol{E}_p}}$ with value of $c = 3$. Next, drop index can be given as:

$$\text{drop-index} = \min/\max_{\forall p \in \mathcal{E}} \Big\{ \sum(\mathcal{A}_{\boldsymbol{E}_p} < \mu_{\mathcal{A}_{\boldsymbol{E}_p}} - 3.0 \times \sigma_{\mathcal{A}_{\boldsymbol{E}_p}}) + \sum(\mathcal{A}_{\boldsymbol{E}_p} > \mu_{\mathcal{A}_{\boldsymbol{E}_p}} + 3.0 \times \sigma_{\mathcal{A}_{\boldsymbol{E}_p}}) \Big\} \tag{13}$$

④ **Expert Activation Norm (EAN):** In this criterion, we calculate the $l2$-norm across the hidden dimension for the accumulated activation ($\mathcal{A}_{\boldsymbol{E}_p}$) of expert $\boldsymbol{E}_p$. Overall activation norm of $\boldsymbol{E}_p$ is estimated as the sum of $l2$-norm over all hidden dimensions and the drop-index is given as:

$$\text{drop-index} = \min/\max_{\forall p \in \mathcal{E}} \big\{ \texttt{sum}(\texttt{norm}_{l2}(\mathcal{A}_{\boldsymbol{E}_p}, \text{dim=0})) \big\} \tag{14}$$

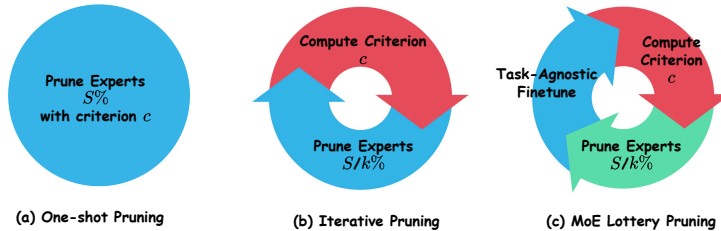

Figure 3: **Overview of Different Expert Pruning Strategies:** Given a target expert sparsity of $S\%$, (a) *One-shot pruning:* removes $S\%$ of experts from each layer $L$ from MoE based on one-time estimation of criterion $c$; (b) *Iterative pruning:* removes $S/k\%$ of experts before re-estimation of criterion $c$ for $k$-rounds; (a) *MoE Lottery pruning:* removes $S/k\%$ of experts followed by *task-agnostic* budget finetuning using calibration data before re-estimation of criterion $c$ for $k$-rounds.

### 2.5 GRADIENT-GUIDED EXPERT IMPORTANCE

① **Expert Gradient Similarity Criterion (EAS):** Given the calibration set of tokens $\mathcal{X}$, we first pass it through the model in batches and accumulate the gradient for all the expert's weight matrices. Consider $\boldsymbol{W}^g_{\boldsymbol{E}_p}$ be the gradient corresponding to the weight matrix of expert $\boldsymbol{E}_p$. We flatten the gradient matrix for all experts of layer $l$ and calculate the pairwise cosine similarity across them.

$$\cos_{n \times n} = \texttt{pairwise-cosine}_{\forall (p,q) \in \mathcal{E} \times \mathcal{E}}(\texttt{flatten}(\boldsymbol{W}^g_{\boldsymbol{E}_p}))$$
$$\text{drop-index} = \texttt{min/max}_{\forall p \in \mathcal{E}}\big\{\texttt{sum}(\cos_{[p,:]}) - \cos_{[p,p]}\big\} \tag{15}$$

② **Expert Gradient Entropy Criterion (EAE):** Gradient entropy is a measurement of information encoded (Guan et al., 2019) within them, and it can be a well-suited indicator for judging the expert importance with the privilege of finetuning. Similar to activation entropy, we estimate gradient entropy by calculating the standard deviation across the hidden dimension of accumulated gradients as:

$$H(\boldsymbol{W}^g_{\boldsymbol{E}_p}) \propto \sum_j \texttt{log}[\sigma(\boldsymbol{W}^{g^j}_{\boldsymbol{E}_p})] \tag{16}$$

③ **Expert Gradient Outliers Criterion (EAO):** In this criterion, we estimate the number of outliers in the accumulated gradients of experts. Given $\boldsymbol{W}^g_{\boldsymbol{E}_p}$ corresponding to weight of expert $\boldsymbol{E}_p$, we count number of outliers outside interval $\mu_{\boldsymbol{W}^g_{\boldsymbol{E}_p}} \pm c \times \sigma_{\boldsymbol{W}^g_{\boldsymbol{E}_p}}$ with value of $c = 3$.

$$\text{drop-index} = \texttt{min/max}_{\forall p \in \mathcal{E}}\big\{\sum(\boldsymbol{W}^g_{\boldsymbol{E}_p} < \mu_{\boldsymbol{W}^g_{\boldsymbol{E}_p}} - 3 \times \sigma_{\boldsymbol{W}^g_{\boldsymbol{E}_p}}) + \sum(\boldsymbol{W}^g_{\boldsymbol{E}_p} > \mu_{\boldsymbol{W}^g_{\boldsymbol{E}_p}} + 3 \times \sigma_{\boldsymbol{W}^g_{\boldsymbol{E}_p}})\big\}$$
$$\tag{17}$$

④ **Expert Gradient Norm Criterion(EAN):** In this criterion, we calculate the $l2$-norm of gradients of weights for all experts of layer $l$ of model $\boldsymbol{M}$. Depending on the $\texttt{min}$ or $\texttt{max}$ argument, we select expert $\boldsymbol{E}_p$ that has min or max gradient norm for dropping.

$$\text{drop-index} = \texttt{min/max}\big\{\text{norm}_{l2}(\boldsymbol{W}^g_{\boldsymbol{E}_p})\big\} \tag{18}$$

## 3 MOE LOTTERY SUBNETWORKS: BLESSING FROM TASK-AGNOSTIC BUDGET FINTETUNING

Expert-level sparsification of SMoEs involves identifying $r$ experts with the least importance using criterions outlined in Section 2 and discarding them to reduce exorbitant memory requirements of loading $n$ experts. Dropping experts require explicit handling of the routing gate function by removing the entry corresponding to dropped experts. In our work, we found that gating function is highly sensitive to any modification and an ad-hoc deletion of $r$ entries from the router matrix (*i.e.,* $\boldsymbol{W}^{d \times n} \rightarrow \boldsymbol{W}^{d \times n-r}$) not only lead to significant performance degradation but also induces heavier load on few among remaining $n - r$ experts. Prior works have limited exploration of *one-shot* removal of $r$ experts to achieve a sparsity ratio of $s\%$ and overlooked attention at finetuning to address the sub-optimal state of SMoE subnetwork after sparsification.

In this work we adopt motivation from the success of lottery ticket hypothesis (Frankle & Carbin, 2018; 2019) and explore: ① *iterative pruning* of experts in $k$-rounds to attain sparsity ratio of

**Expert Weight-Based Pruning (PPL on C4)**

| Criterion (c) | 12.5% | 25.0% | 37.5% | 50.0% | 62.5% | 75.0% |
|---|---|---|---|---|---|---|
| Max-EWS | 7.98 | 8.56 | 9.41 | 10.41 | 11.93 | 14.91 |
| Min-EWS | 10.34 | 10.82 | 11.60 | 13.21 | 16.12 | 19.71 |
| Max-RWN | 8.13 | 8.72 | 9.47 | 10.13 | 11.29 | 13.79 |
| Min-RWN | 10.44 | 10.98 | 11.68 | 13.60 | 16.76 | 31.44 |
| Max-WSR | 7.89 | 8.37 | 8.93 | 10.80 | 12.98 | 15.28 |
| Min-WSR | 10.93 | 11.52 | 13.10 | 16.20 | 22.46 | 26.86 |
| Max-EWN | 7.91 | 8.46 | 9.12 | 9.96 | 11.04 | 17.63 |
| Min-EWN | 10.68 | 11.02 | 12.83 | 14.24 | 17.47 | 29.29 |

% Experts Dropped

**Expert Behaviour-Based Pruning (PPL on C4)**

| Criterion (c) | 12.5% | 25.0% | 37.5% | 50.0% | 62.5% | 75.0% |
|---|---|---|---|---|---|---|
| Max-EUF | 10.28 | 10.29 | 11.31 | 13.23 | 13.64 | 22.38 |
| Min-EUF | 8.21 | 8.81 | 9.22 | 13.47 | 16.63 | 21.64 |
| Max-ECP | 10.62 | 10.90 | 11.70 | 13.30 | 15.98 | 25.54 |
| Min-ECP | 7.96 | 8.47 | 9.12 | 10.50 | 11.92 | 24.89 |
| Max-EVC | 8.01 | 8.64 | 14.55 | 17.01 | 21.70 | 34.65 |
| Min-EVC | 7.96 | 8.69 | 9.41 | 10.18 | 16.42 | 20.99 |
| Max-ETS | 7.98 | 8.46 | 9.09 | 10.13 | 11.72 | 14.15 |
| Min-ETS | 7.98 | 8.73 | 9.45 | 10.33 | 11.60 | 21.30 |

% Experts Dropped

**Expert Activation-Based Pruning (PPL on C4)**

| Criterion (c) | 12.5% | 25.0% | 37.5% | 50.0% | 62.5% | 75.0% |
|---|---|---|---|---|---|---|
| Max-EAS | 7.95 | 8.57 | 9.28 | 10.25 | 11.68 | 14.99 |
| Min-EAS | 10.90 | 11.05 | 11.67 | 13.70 | 16.97 | 21.90 |
| Max-EAE | 11.14 | 12.64 | 13.82 | 16.43 | 19.80 | 26.76 |
| Min-EAE | 7.89 | 8.40 | 8.99 | 9.72 | 11.06 | 13.36 |
| Max-EAO | 7.93 | 8.66 | 9.57 | 10.64 | 12.07 | 15.53 |
| Min-EAO | 10.96 | 11.99 | 12.82 | 14.57 | 17.45 | 23.20 |
| Max-EAN | 11.25 | 12.75 | 14.23 | 16.99 | 20.75 | 28.99 |
| Min-EAN | 7.89 | 8.38 | 9.00 | 9.76 | 11.00 | 13.05 |

% Experts Dropped

**Expert Gradient-Based Pruning (PPL on C4)**

| Criterion (c) | 12.5% | 25.0% | 37.5% | 50.0% | 62.5% | 75.0% |
|---|---|---|---|---|---|---|
| Max-EGS | 7.95 | 8.57 | 9.26 | 10.13 | 11.36 | 14.01 |
| Min-EGS | 10.90 | 11.05 | 11.67 | 13.68 | 17.03 | 22.05 |
| Max-EGE | 8.06 | 11.80 | 13.52 | 15.76 | 20.26 | 32.98 |
| Min-EGE | 7.88 | 8.26 | 9.09 | 9.88 | 11.00 | 13.09 |
| Max-EGO | 8.21 | 8.74 | 9.43 | 10.39 | 12.03 | 14.79 |
| Min-EGO | 10.58 | 12.18 | 13.74 | 16.47 | 21.14 | 32.31 |
| Max-EGN | 8.05 | 8.69 | 9.38 | 10.34 | 12.13 | 15.92 |
| Min-EGN | 10.34 | 10.96 | 12.55 | 14.33 | 18.44 | 24.51 |

% Experts Dropped

Figure 4: Performance comparison (perplexity on C4) of `Mixtral-8×7B Base` Lottery Subnetworks identified by dropping experts iteratively using various criterions from MC-Suite. Original `Mixtral-8×7B Base` checkpoint achieves 7.44 perplexity on C4 validation set. *Min & Max* represents an expert ($e$) with minimum/maximum score of a criterion ($c$) in a MoE layer $l$ is dropped.

| Criterion ($c$) | 12.5% | 25.0% | 37.5% | 50.0% | 62.5% | 75.0% |
|---|---|---|---|---|---|---|
| Random Dropping (One-shot) | 9.01 | 11.02 | 11.95 | 15.21 | 21.10 | 34.47 |
| Random Dropping (Iterative) | 9.78 | 11.12 | 13.06 | 15.46 | 22.76 | 38.94 |
| Random Dropping (w. MoE Lottery) | 9.66 | 10.54 | 11.83 | 13.71 | 18.23 | 33.05 |
| Max-Router Weight Norm (RWN) | 8.47 | 9.00 | 9.87 | 10.70 | 13.50 | 17.26 |
| Max-Expert Token Similarity (ETS) | 8.28 | 8.82 | 9.50 | 10.43 | 12.48 | 16.03 |
| Min-Expert Gradient Entropy (EGE) | 8.17 | 8.84 | 9.54 | 10.45 | 11.88 | 15.08 |
| Min-Expert Activation Norm (EAN) | 8.18 | 8.63 | 9.21 | 9.99 | 11.43 | 14.02 |

Table 1: Performance comparison (perplexity on C4) of `Mixtral-8×7B Instruct` Lottery Subnetworks identified by various top-performing criterions from MC-Suite. Original `Mixtral-8×7B Instruct` checkpoint achieves 7.82 perplexity on C4 validation set.

$s\%$; ② incorporation of task-agnostic finetuning on next token prediction task to stabilize the sub-optimal state of SMoE subnetworks. Moreover, an iterative pruning strategy with *re-estimation* of importance criterion enables taking into account the impact of thee removal of the first round of experts on deciding the importance of remaining experts. We propose **MoE Lottery Subnetwork**, which relies on iterative *estimate-prune-finetune* procedure as shown in Figure 3. Note that we choose to state *budget finetuning* because we found that one doesn't require extensive finetuning iterations but a marginal amount is sufficient to obtain desirable performance gains (Appendix A.2).

Our experimental results in this section have two-folds. *Firstly,* we perform a comprehensive evaluation of the criterions of MC-Suite (Section 2) using MoE lottery subnetworks with varying sparsity ratios of $s \in \{12.5\%, ..., 75.0\%\}$. *Secondly,* we aim to understand the merits of iterative pruning and task-agnostic budget finetuning by selecting top-performing MC-Suite criterions.

### 3.1 MC-SUITE AND MOE LOTTERY SUBNETWORKS

MC-Suite consists of a series of criterions from four diverse perspectives that provide "clues" for identifying experts that contribute least to the original SMoE model and thus can be discarded. Given a criterion $c$ from MC-Suite, we study both maximizing and minimizing $c$ while generating the MoE lottery subnetworks to understand the characteristics of retained experts and its impact on the final performance. Figure 4 presents the `C4` validation perplexity of MoE lottery subnetworks of Mixtral-8×7B `Base` model where an expert $e$ from a MoE layer $l$ is dropped subjected to maximum or minimum value of $c$ across other fellow experts in $l$. Table 1 presents the comparison of best-performing criterions from four different perspectives of MC-Suite along with randomly selected expert dropping baseline. It can be clearly observed that the usage of criterions from MC-Suite significantly helps in improving the performance of MoE lottery subnetworks. In our experimental setting, we choose to drop 32 experts (*i.e.,* 12.5% sparsity) in every round of iterative pruning with one expert per layer. Our experiments found that a non-uniform dropping of experts per layer by estimating $c$ globally creates bottleneck layers, with some layers having significantly high sparsity while some remain unpruned, leading to diminished finetuning benefits and sharding simplicity.

| % Experts Dropped | Random Dropping | | | Min-Activation Norm (Min-EAN) | | | Min-Gradient Entropy (Min-EGE) | | |
|---|---|---|---|---|---|---|---|---|---|
| | One-shot | Iterative | MoE Lottery | One-shot | Iterative | MoE Lottery | One-shot | Iterative | MoE Lottery |
| 0% | | | | | 7.44 | | | | |
| 12.5% | 11.25 | 7.94 | 7.89 | 7.95 | 7.90 | 7.89 | 7.89 | 7.89 | 7.88 |
| 25.0% | 12.74 | 10.98 | 11.01 | 8.56 | 8.53 | 8.38 | 8.47 | 8.41 | 8.26 |
| 37.5% | 13.89 | 13.19 | 12.22 | 12.87 | 9.35 | 9.00 | 13.33 | 9.48 | 9.09 |
| 50.0% | 17.08 | 15.85 | 14.13 | 14.74 | 10.44 | 9.76 | 15.37 | 10.72 | 9.88 |
| 62.5% | 30.41 | 18.79 | 20.60 | 21.36 | 12.55 | 11.00 | 22.21 | 12.81 | 11.00 |
| 75.0% | 36.92 | 32.73 | 27.33 | 30.59 | 17.39 | 13.05 | 35.83 | 17.70 | 13.09 |

Table 2: **Improved Language Modelling Abilities:** Performance comparison of MoE Lottery Sub-networks identified using criterion ($c$) with respect to Iterative and One-shot pruning. MoE Lottery Subnetworks, which are supplemented with task-agnostic finetuning, are able to restore a better optimal state impacted by ad-hoc derivation from their dense counterpart.

| | Criterion ($c$)= Min-Activation Norm | 0% | 12.5% | 25% | 37.5% | 50% | 62.5% | 75% |
|---|---|---|---|---|---|---|---|---|
| MMLU | One-shot Pruning | | 52.97 | 43.97 | 13.55 | 18.91 | 12.63 | 5.82 |
| | Iterative Pruning | 60.01 | 48.51 | 47.81 | 45.63 | 35..74 | 29.71 | 23.88 |
| | MoE Lottery Networks | | 49.54 | 49.65 | 47.13 | 40.79 | 37.24 | 28.12 |
| WinoGrande | One-shot Pruning | | 55.13 | 50.09 | 37.45 | 36.91 | 20.44 | 24.63 |
| | Iterative Pruning | 56.59 | 55.90 | 52.17 | 49.96 | 48.53 | 47.11 | 50.35 |
| | MoE Lottery Networks | | 55.92 | 52.98 | 50.96 | 49.56 | 49.30 | 50.74 |

Table 3: **Improved Zero-shot Downstream Performance:** Downstream task performance comparison of MoE Lottery Subnetworks identified using criterion ($c$) with respect to Iterative and One-shot pruning in zero-shot setting (no in-context examples). MoE Lottery networks tend to have superior abilities to follow instructions required to complete the downstream tasks.

The benefits of MC-Suite are **not** limited to exploration of the best recipe to identify least important experts for dropping, but extends in deriving many valuable hidden insights of important experts. We comprehend few interesting findings as: ① activation and gradient-guided criterions (minimum activation norm and gradient entropy) that take into account both input tokens and model parameters achieves the *superior performance* over conventional criterions such as expert usage, expert weight similarity, *etc.*; ② surprisingly, $l2$-norm of router weight matrix turn out to be the best performing candidate in comparison to other expert weight based criterions; ③ dropping experts with higher vocabulary coverage lead to a significant drop in performance which indicate efforts to improve specialization across experts in MoEs can be non-conducive for expert-level sparsification; ④ dominant experts tends to have *lower stable-rank*, which aligns with recent findings of (Jaiswal et al., 2024; Zhang et al., 2024) that LLMs weight matrices which are critical and well-trained also have comparatively lower stable-rank with further compression potential with orthogonal techniques like low-rank factorization; ⑤ our **novel** criterion of *entropy quantification of activation and gradient* aiming to measures information encoded within them, turns out to best performing recipes for estimating expert importance and also favourable for downstream task finetuning. Interestingly, while comparing the impact of expert-level sparsification for Mixtral-8×7B `Base` and `Instruct` check-points, we found that task-agnostic finetuning has comparatively lower benefits for `Instruct` in comparison to `Base` model suggesting to perform expert dropping before instruction tuning.

## 3.2 Understanding the Merits of Task-Agnostic Budget Finetuning

In this section, we attempt to unveil the true merits of the iterative *estimate-prune-finetune* procedure of MoE lottery subnetworks. To investigate the benefits contributed by iterative pruning and task-agnostic finetuning, we present performance comparison for one-shot, iterative pruning, and MoE lottery subnetworks. *Firstly*, Table 2 illustrate the *improved language modelling abilities* measured using validation perplexity of C4 dataset where MoE lottery networks (with Min-EAN and Min-EGE criterion) can achieve $\sim 3\times$ better performance compared to one-shot pruning, while iterative pruning without any finetuning can still achieve $\sim 2\times$ superior performance. It is also interesting to note that even the random expert selection baseline significantly benefits from iterative pruning and finetuning with $\sim 9.5$ points better perplexity than one-shot pruning. *Secondly,* Table 3 presents the *improved zero-shot downstream performance* (no in-context examples) of MoE lottery subnetworks over one-shot and iterative pruning at varying sparsity levels on MMLU and WinoGrande. Clearly, it can be observed that while one-shot pruning starts performing worse than random guess with merely a 25% sparsity ratio; MoE lottery networks performance doesn't drop below random guess even at non-trivial sparsity ratio (62.5%-75.0%). Moreover, the the contribution of iterative *estimate-prune-finetune* become more notable with increasing sparsity ratios.

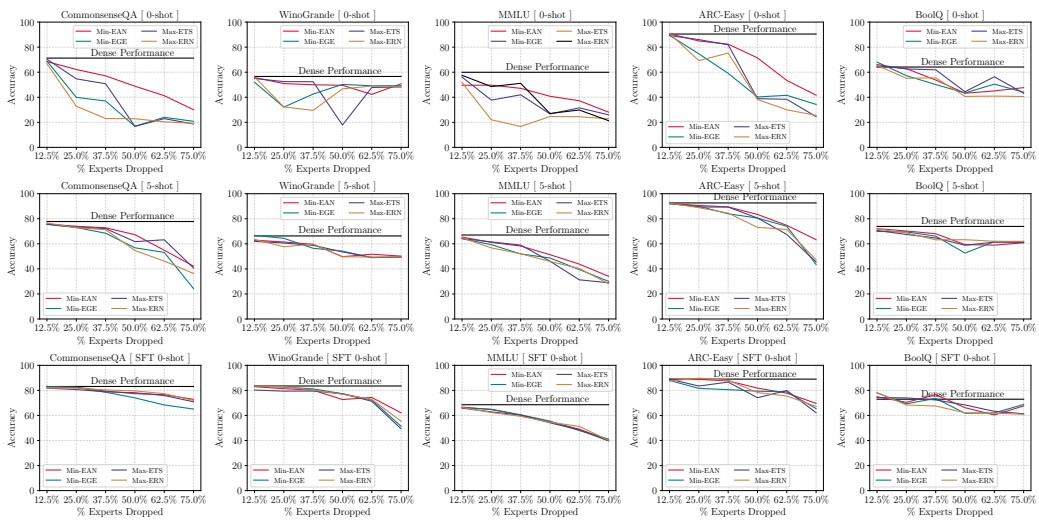

Figure 6: Downstream task performance of MoE Lottery Subnetworks at varying sparsity level when augmented with external instruction following capabilities using k-shot examples (Row 2) and supervised finetuning (Row 3) using instruction-tuning dataset.

Next, we ask an interesting question: *How does task-agnostic finetuning, which aims to re-adjust the router weight, influence the load distribution across experts?* To this end, Figure 5 illustrates the expert load distribution[7] of remaining experts of a MoE layer from Mixtral-8×7B `Base` model with 50% expert sparsity ratio before (dashed red line) and after (solid green line) task-agnostic finetuning using $C4$ dataset. It can be clearly observed that our proposed finetuning sub-

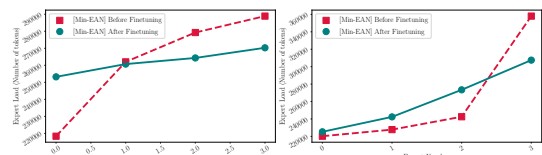

Figure 5: Improved load balancing across experts ($l = 6$ & 30) for Mixtral-8×7B `Base` model before and after task-agnostic finetuning with $C4$.

routine can significantly help in induced skewness in load distribution across experts due to expert droping and removal of its entry from the router gating function. Note that a well-balanced load distribution across experts is encouraged to facilitate better GPU memory utilization and speedup.

## 4 WHAT IS LOST V/S WHAT PREVAILS? AN IN-DEPTH INVESTIGATION OF EXPERT DROPPING AND LOST CAPABILITIES

SMoE models require enormous memory to host experts during inference while being known to have poor utilization of its capacity. In recent times, multiple LLM compression techniques (*e.g.,* weight sparsity, quantization, low-rank factorization, etc.) are being developed to address the memory and computational bottleneck. Some works (Jaiswal et al., 2023a; Hong et al., 2024; Yin et al., 2023a) attempt to understand the impact of compression on pretrained checkpoints while handling knowledge-intensive tasks, trust, and safety. Motivated by their findings, we aim to understand the impact of dropping least important and redundant experts during expert-level sparsification of SMoEs. Given that SMoEs are trained using a Top-$k$ routing policy, each token is processed by $k$ experts, promoting redundancy and less sensitivity to expert dropping by design choice. We ask: *What capabilities of full-SMoEs are severely impacted by the removal of least dominant experts?*

At first, a *narrow view* of the zero-shot downstream evaluation of SMoE subnetworks with expert-level sparsification indicates a sharp performance drop compared to the full-SMoEs. Figure 6 (row 1) illustrates the zero-shot performance of MoE lottery subnetworks identified with four criterions from MC-Suite on 5 popular reasoning and knowledge-intensive tasks. It can be clearly observed that the expert-dropping tends to have an acute impact on the downstream tasks but we pause and ask: *Is this abrupt performance degradation incurring due to loss of pretraining knowledge and reasoning abilities or instruction-following abilities?* We **conjecture** that when we drop the least

---

[7]Expert ($e$) Load: Given a fixed number of input tokens, # tokens processed by the expert $e$.

dominant experts, SMoEs instruction following capabilities are predominantly hurt, and it can be restored to a notable extent with external support.

To experimentally validate our conjecture, we design the controlled experiments in three folds: ① *zero-shot setting* which directly evaluate pruned SMoE performance on downstream tasks without any in-context example; ② *k-shot setting* which provide $k$ in-context examples as external assistance for compressed LLMs to follow downstream instructions; ③ *supervised finetuning (SFT)* that aim to explicitly embed external instruction following support in compressed SMoE checkpoint by fin-tuning using instruction following dataset. Figure 6 (row 2 & 3) illustrates that external instruction-following support can *impressively minimize* the performance gap due to expert-level sparsification on complex reasoning downstream tasks. Note that for fair comparsion, our full-SMoE baselines represented as straight lines are also provided exactly similar external instruction-following support. Interestingly, we can observe that SFT, even with the zero-shot setting, can enable **robust** performance of compressed SMoE models at non-trivial sparsity ratios ($\geq 50\%$). Moreover, for some comparatively easier tasks (*e.g.,* BoolQ, ARC-easy), it facilitates pruned SMoEs to outperform the full-SMoE baseline.

## 5 EXPERT DROPPING V/S LLM WEIGHT PRUNING TECHNIQUES

LLM weight pruning algorithm (Yin et al., 2023b; Jaiswal et al., 2023b; Sun et al., 2023; Frantar & Alistarh, 2023) involves removing non-significant weights parameters by setting them to zero. Recent hardware advancements have enabled practical speedup for structural N:M sparsity patterns (Nvidia, 2020; Zhou et al., 2021). In this section, we investigate the downstream task performance of the expert-level sparsification method with the representative weight pruning baselines (random, magnitude, and wanda). For expert-level sparsification, we present MoE lottery networks with random and minimum activation norm criterions to identify dominant experts. Provided the hardware supported $2 : 4$ weight sparsity patterns, we choose expert drop ratio ($r = 4$) per layer to achieve 50% sparsification for both categories for fair comparison.

| Model | Method | Sparsity | Arc-c | ARC-e | HellaSwag | MMLU | WinoGrande | Average |
|---|---|---|---|---|---|---|---|---|
| **Mixtral 8×7B** | None | $r = 8$ | 78.18 | 91.94 | 64.88 | 60.01 | 56.59 | 70.32 |
| | Random Pruning | $2 : 4$ | 19.47 | 48.90 | 28.90 | 17.05 | 22.07 | 27.27 |
| | Magnitude Pruning | $2 : 4$ | 31.07 | 69.76 | 43.23 | 42.77 | 38.56 | 45.07 |
| | Wanda Pruning | $2 : 4$ | 43.82 | 70.16 | 53.16 | 50.21 | 48.96 | 52.91 |
| | Min-EAN Expert Pruning | $r = 4$ | 60.02 | 71.41 | 50.78 | 51.33 | 49.56 | 56.62 |
| **Mixtral 8×7B Instruct** | None | $r = 8$ | 81.86 | 93.21 | 78.06 | 64.67 | 63.77 | 76.31 |
| | Random Pruning | $2 : 4$ | 23.68 | 56.42 | 37.01 | 22.15 | 29.07 | 31.94 |
| | Magnitude Pruning | $2 : 4$ | 54.96 | 69.44 | 57.18 | 29.08 | 40.79 | 50.29 |
| | Wanda Pruning | $2 : 4$ | 61.92 | 80.23 | 62.90 | 51.05 | 55.30 | 62.28 |
| | Min-EAN Expert Pruning | $r = 4$ | 68.50 | 83.59 | 64.46 | 48.56 | 54.65 | 63.95 |

Table 4: **Expert-level Sparsification V/s LLM Weight Pruning:** Downstream task performance comparison in zero-shot setting (no in-context example) of Mixtral 8×7B `base` and `Instruct` when compressed using expert-level sparsification techniques v/s SoTA LLM pruning methods.

Table 4 summarizes the performance comparison in zero-shot setting for all baselines and MoE Lottery subnetwork for Mixtral-8×7B `Base` and `Instruct` checkpoints. It can be observed that expert-level sparsification can achieve $\sim 3.6\%$ average performance gain over the Wanda pruning while a notable $\sim 16.2\%$ imporvement on ARC-c downstream task. In addition, we also find that the performance benefits for `Base` model is comparatively superior than `Instruct` suggesting it is favourable to perform expert-level sparsification on the `Base` model before instruction tuning.

## 6 CONCLUSION

In this paper, we provide a detailed investigation of multiple expert importance estimation techniques (MC-Suite) to identify the best recipe for selecting the least knowledgeable experts that can be dropped without sacrificing the vital knowledge and capabilities of the SMoE. We propose to adopt a iterative pruning strategy with task-agnostic finetuning as a correction measure to minimize the drastic impact on SMoE capabilities. We present and experimentally validate an interesting con-jecture that during expert dropping, SMoE instruction following capabilities are predominantly hurt, and SMoE performance can be notably recovered with a few-shot demonstration or supervised fine-tuning. In our future work, we plan to investigate and disentangle the instruction-following abilities and pretraining knowledge across the parameters of SMoE experts.

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

## A APPENDIX

### A.1 RELATED WORK

**SMoE and Its Superiority.** It is widely acknowledged that scaling model size benefits performance by enhancing learning capacity and generalization ability (Brown et al., 2020; Kaplan et al., 2020). To achieve more efficient model scaling, Sparsely activated Mixture-of-Experts (SMoE) (Shazeer et al., 2017b; Zoph et al., 2022; Du et al., 2022) has emerged as a widely adopted approach, enabling the training of larger models with negligible additional computational overhead (Jiang et al., 2024; Dai et al., 2024; DeepSeek-AI et al., 2024). Given the predominance of Transformer architectures in NLP, numerous research efforts have focused on incorporating MoE layers within the feed-forward neural networks of these models. In pursuit of enhanced SMoE models, various iterations of the standard MoE architecture have been proposed. For example, DeepSeek-MoE (Dai et al., 2024; DeepSeek-AI et al., 2024) utilizes a large number of finely segmented experts, designating a subset as shared experts to capture common knowledge. More recently, Mixtral (Jiang et al., 2024) has demonstrated that SMoE can achieve performance comparable to full-parameter LLMs while utilizing significantly fewer active parameters.

**Compression for LLMs and SMoEs.** LLMs have demonstrated remarkable success. However, their substantial memory and computational requirements pose deployment challenges. Numerous model compression techniques have been proposed to address this issue. Algorithmically, these methods can be classified into three main categories: ① Quantization, which converts float32 weights or activations to lower-bit representations(Lin et al., 2023; Frantar et al., 2022; Jaiswal et al., 2022; Xiao et al., 2024); ② Pruning, which eliminates less critical components, such as weights, neurons, or layers (LeCun et al., 1989; Han et al., 2016; Sun et al., 2023); ③ Knowledge distillation, which transfers knowledge from a larger model to a smaller one (Gou et al., 2021; Li et al., 2024b; Rajbhandari et al., 2022). In this study, we concentrate on model pruning for compression, which is generally divided into *structured* and *unstructured* approaches. Structured pruning methods (Liu et al., 2017; Molchanov et al., 2019; Shen et al., 2022; Fang et al., 2023) eliminate entire structured components of a network, facilitating straightforward GPU acceleration. Existing techniques primarily rely on weight or activation statistics of neurons (Dubey et al., 2018; Molchanov et al., 2017). Unstructured methods (Han et al., 2015; Paul et al., 2022; Hoang et al., 2023) operate at the individual weight level, preserving performance at higher sparsity levels but typically requiring additional effort to enable GPU speedups (Mishra et al., 2021).

SMoE architectures enable the scaling of LLMs but necessitate substantial memory to host experts while exhibiting expert redundancy. To address these challenges, numerous studies have also focused on developing SMoE-model-specific compression techniques. Initial approaches (Chen et al., 2022; Kim et al., 2021; Koishekenov et al., 2023; Sarkar et al., 2024) propose expert pruning based on utilization metrics; however, these methods often resulted in diminished performance. Subsequent research (Rajbhandari et al., 2022; Fedus et al., 2022; Artetxe et al., 2022) explores the creation of smaller models, either dense or SMoE-based, with reduced layer counts through knowledge distillation (KD). While effective, this approach demands significant computational resources and fails to address the inherent redundancy among experts. More recently, MC-SMoE (Li et al., 2024b) dynamically merges experts during inference time, though it is limited to specific tasks. Besides pruning-based methods, there are also a few works that specifically study quantization in SMoE models Li et al. (2024a).

### A.2 TRAINING DURATION AND MOE LOTTERY NETWORKS

MoE lottery subnetworks rely on *estimate-prune-finetune* procedure to mitigate the abrupt impact of expert dropping of the resultant subnetwork. More specifically, finetuning routine using pre-training objectives helps in balancing expert load distribution and performance improvement. One natural question that arises is: *Given the enormous computational cost of finetuning SMoEs, how much finetuning will be sufficient to achieve a reasonable performance gain facilitated by it?*

Table 5 presents the performance (perplexity) of Mixtral-7×8B `Base` and `Instruct` model checkpoints when 6 out of 8 experts are dropped from every layer using the Minimum Expert Activation Norm (Min-EAN) criterion. Each column in Table 5 indicates the total number of training tokens used during the finetuning subroutine of the MoE Lottery Subnetwork. It can be clearly observed

| Training Tokens → | 0.25M | 0.51M | 1.13M | 2.27M |
|---|---|---|---|---|
| Mixtral 8×7B | 13.55 | 13.51 | 13.05 | 13.01 |
| Mixtral 8×7B Instruct | 14.82 | 14.19 | 14.02 | 14.08 |

Table 5: Performance comparison (perplexity) wrt. total training tokens used in task-agnostic fine-tuning of Mistral checkpoints with 75% expert dropping.

that the benefits of task-agnostic finetuning saturates after a certain amount of training tokens. More specifically, we found that ∼1 million training tokens are sufficient to address the abrupt impact created by expert dropping and any additional finetuning brings marginal or no gain in performance.

## A.3 UNDERSTANDING EXPERT DROPPING PATTERN ACROSS ONE-SHOT, ITERATIVE & MoE LOTTERY PRUNING

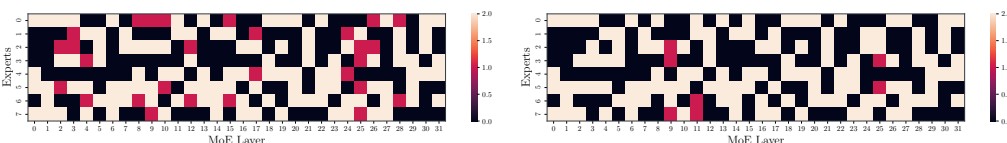

Figure 7: **Dropped Experts Distribution with 50% Sparsity:** (a) Difference of experts identified to be dropped with *one-shot pruning* in comparison with *moe-lottery pruning*, (b) Difference of experts identified to be dropped with *iterative pruning* in comparison with *moe-lottery pruning*. *Light Bisque* color corresponding to an expert ($e_L^i$) indicates agreement across both pruning techniques to drop $e_L^i$, *Dark pink* indicates disagreement to drop, while *Black* indicates agreement to retain $e_L^i$.

In this section, we study the divergence of the selection of experts for pruning of *one-shot* and *iterative pruning* w.r.t. *MoE lottery pruning*. The primary aim of this study is to highlight the benefits of iterative pruning with re-estimation of expert importance criterions. It can be clearly observed from Figure 7(a) that there exists a *significantly high disagreement* (dark pink) between one-shot and iterative pruning while selecting least dominant experts *leading to completely different resultant subnetworks*. The substandard performance of the one-shot method indicates that the identified subnetwork is not of high quality in comparison to iterative pruning. On the other hand, Figure 7(b) illustrates a notable high agreement across experts, which undergoes dropping to achieve a sparsity ratio of 50%. This leads to an interesting conclusion that task-agnostic finetuning does not significantly alter the expert selection choice selection but instead helps in addressing the impact incurred due to sparsification in the form of load imbalance and abrupt performance drop.

## A.4 ADDITIONAL EXPERIMENTAL SETUP

| Hyperparameter | CommonsenseQA | WinoGrande | MMLU | ARC-Easy | BoolQ |
|---|---|---|---|---|---|
| Train Samples (avg. words) | 9741(28.00) | 63238 (39.96) | 1531 (84.97) | 2247 (48.16) | 9427 (14.81) |
| Test Samples (avg. words) | 1221(27.75) | 1267(40.20) | 14042 (84.28) | 2372 (48.42) | 3270 (14.70) |
| Batch Size | 8 | 8 | 4 | 8 | 8 |
| Max_length | 512 | 512 | 512 | 512 | 512 |
| Training Steps | 2500 | 2500 | 1000 | 1500 | 2500 |
| Learning Rate | 0.0001 | 0.0001 | 0.0001 | 0.0001 | 0.0001 |

Table 6: Hyperparamters settings for zero-shot downstream finetuning of Mistral-8×7B models.

Our experiments are conducted on Mixtral MoE `Base` and `Instruct` downloaded from Hug-gingFace. For activation and gradient criterions, we propose to use a task-agnostic calibration $C4$ validation set of 256 samples with `max_seq_len` of 2048. As suggested in Table 5, the benefits of task-agnostic finetuning saturates with no significant benefits of prolonged finetuning, we propose a progressive scheduler for number of training tokens required for $k$ rounds of MoE lottery pruning to miminize compute requirements. More specifically, we double the amount of tokens every round starting from 0.2M tokens for first round. We used `adamw` with a `cosine` learning scheduler

with maximum learning rate of $1e - 6$. With the availability of 8×A100, we use a batch size of 8 and every round we reset the optimizer. Additional details for our downstream finetuning tasks are provided in Table 6 and we followed the exactly same settings for all compression level.

## A.5 PERFORMANCE COMPARISON WITH SoTA MoE EXPERT PRUNING METHODS

| Method | Total Expert Sparsity($\uparrow$) | Accuracy Drop from Dense ($\downarrow$) | Memory Usage($\downarrow$) | Speedup($\uparrow$) |
|---|---|---|---|---|
| Dense | 0 | 0 | ×1 | ×1 |
| Random | 50% | 20.46 | ×0.55 | ×1.27 |
| Lu et al. (2024) | 50% | 14.38 | ×0.55 | ×1.27 |
| Muzio et al. (2024) | 50% | 13.78 | ×0.55 | ×1.27 |
| Ours | 50% | 13.05 | ×0.55 | ×1.27 |

Table 7: Comparison with baseline approaches. MC-Suite Criterion (Min-EAN) achieves the minimal accuracy drop from the dense baseline at all expert sparsity levels. For Muzio et al. (2024) we use the numbers reported in the paper due to unavailability of code to reproduce.

## A.6 MoE EXPERTS AND MC-SUITE CRITERIONS

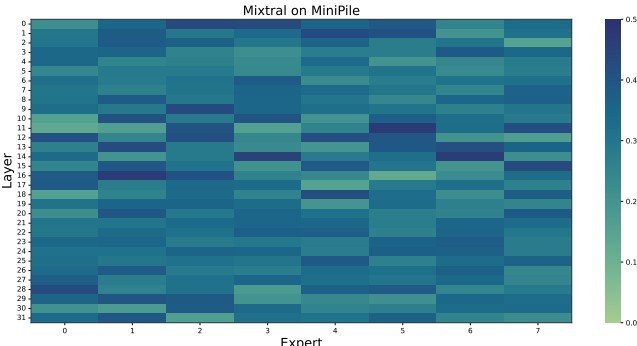

Figure 8: **Experts Vocabulary Coverage Criterion (EVC):** Illustration of experts vocabulary coverage corresponding to different MoE layers from Mixtral-8×7B `Base` model. Experts with minimum vocabulary coverage are better candidates for dropping.

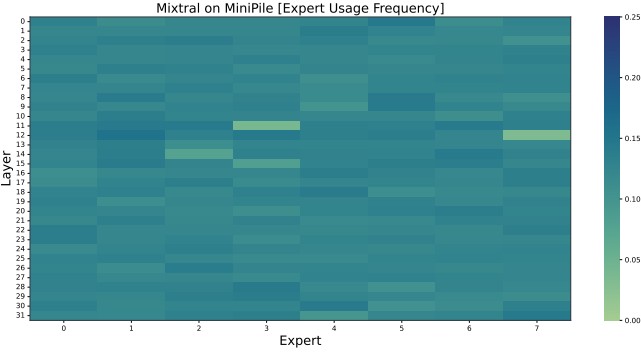

Figure 9: **Experts-Usage Frequency (EUF):** Expert usage frequency indicate how frequently an expert $e$ is activated and above heatmap indicate experts from different MoE layers from Mixtral-8×7B `Base` model. Interestingly, it can be observed that there multiple experts with significantly low expert usage making them good candidate for expert dropping.

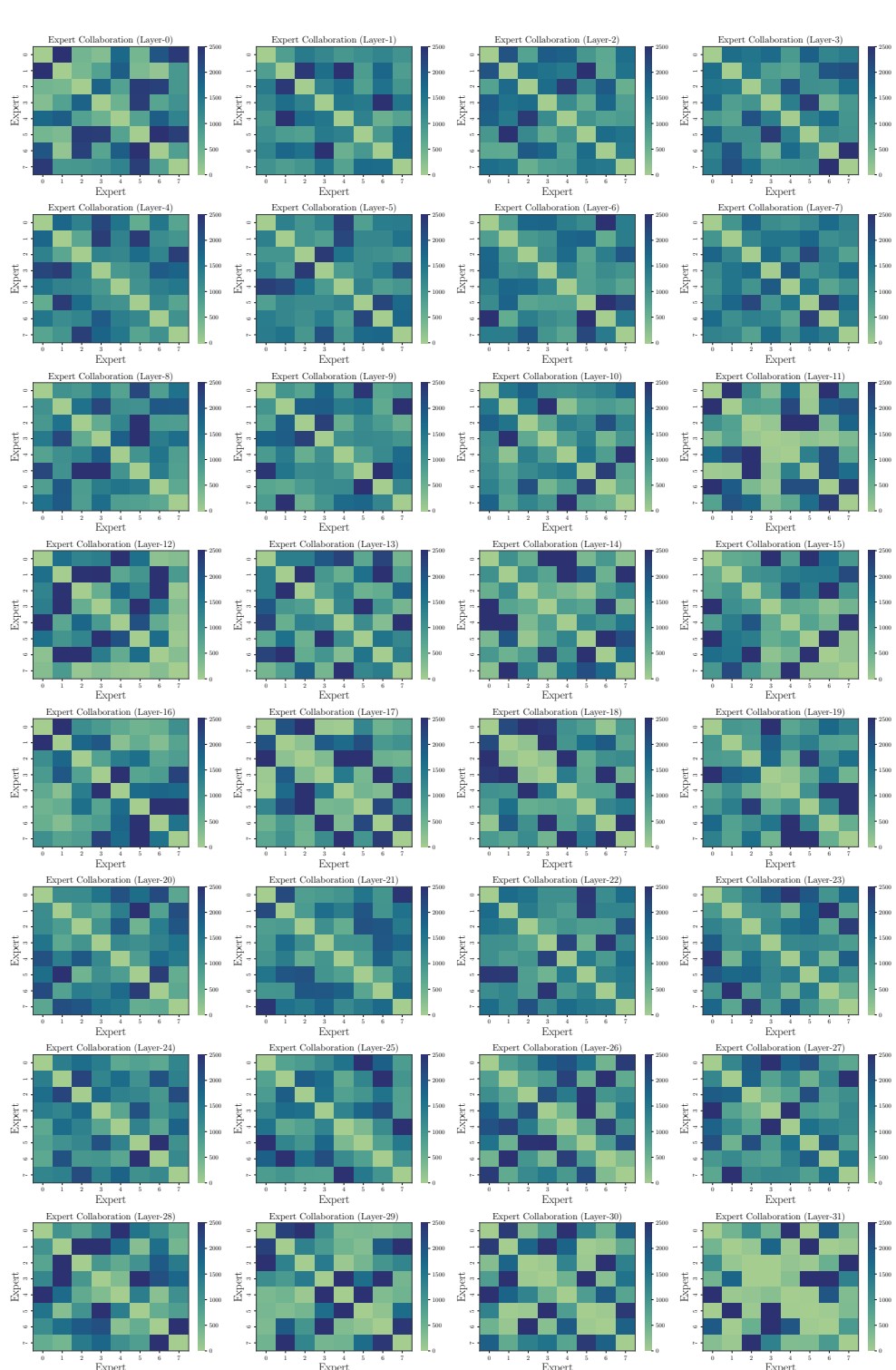

Figure 10: **Experts-Expert Collaboration (ECC):** Snapshot of Expert-Expert Collaboration estimated using C4 dataset for Mixtral-8×7B Base model. Least dominant expert are identified as expert which have highest collaboration with rest of other experts within corresponding layer.

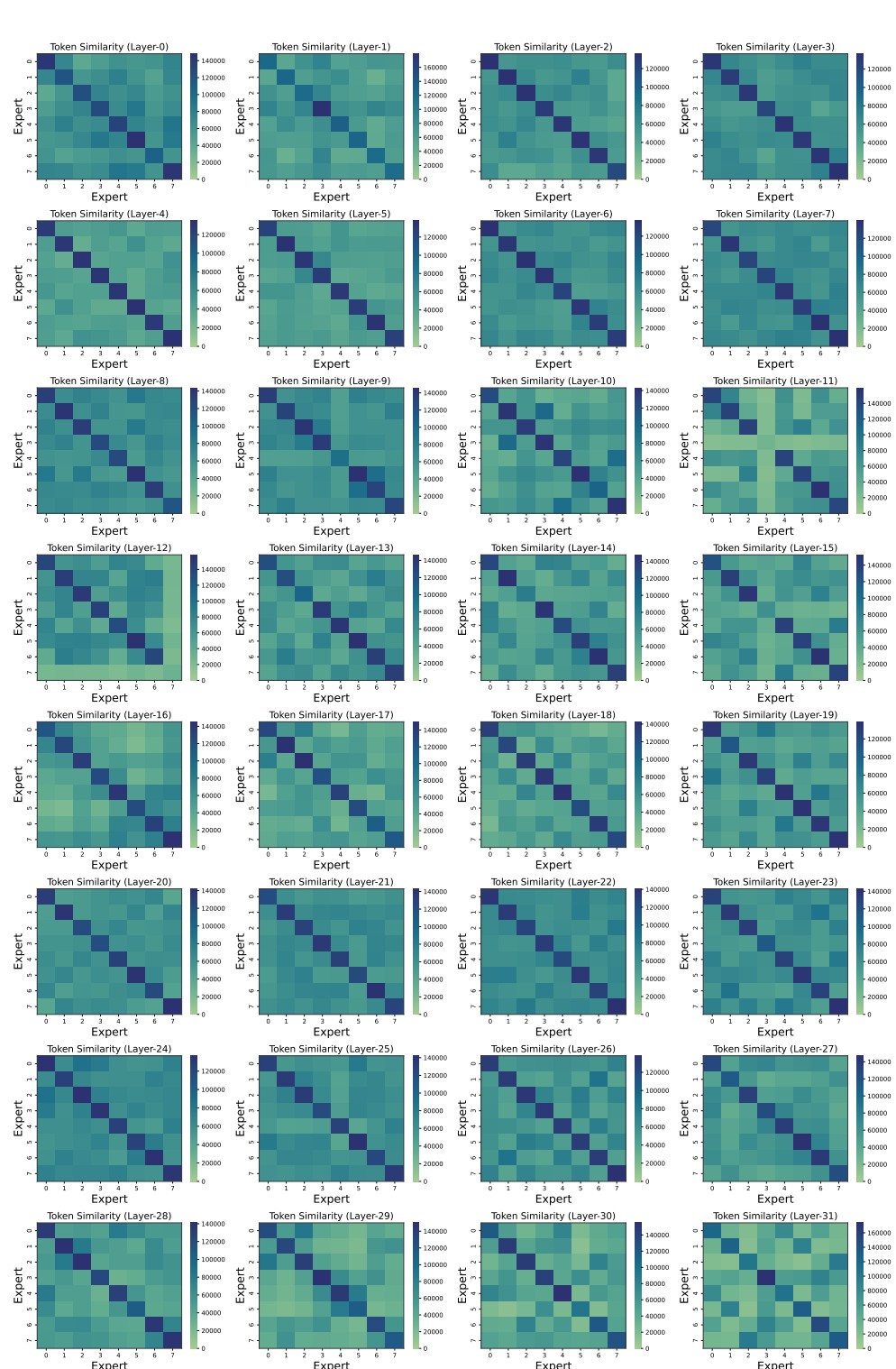

Figure 11: **Expert Input Token Similarity (ETS):** Snapshot of Expert-Expert Input token similarity estimated using C4 dataset for Mixtral-8×7B `Base` model. Higher level of input token similarity indicate existence of redundancy and can be used as a signal to identify least dominant expert.

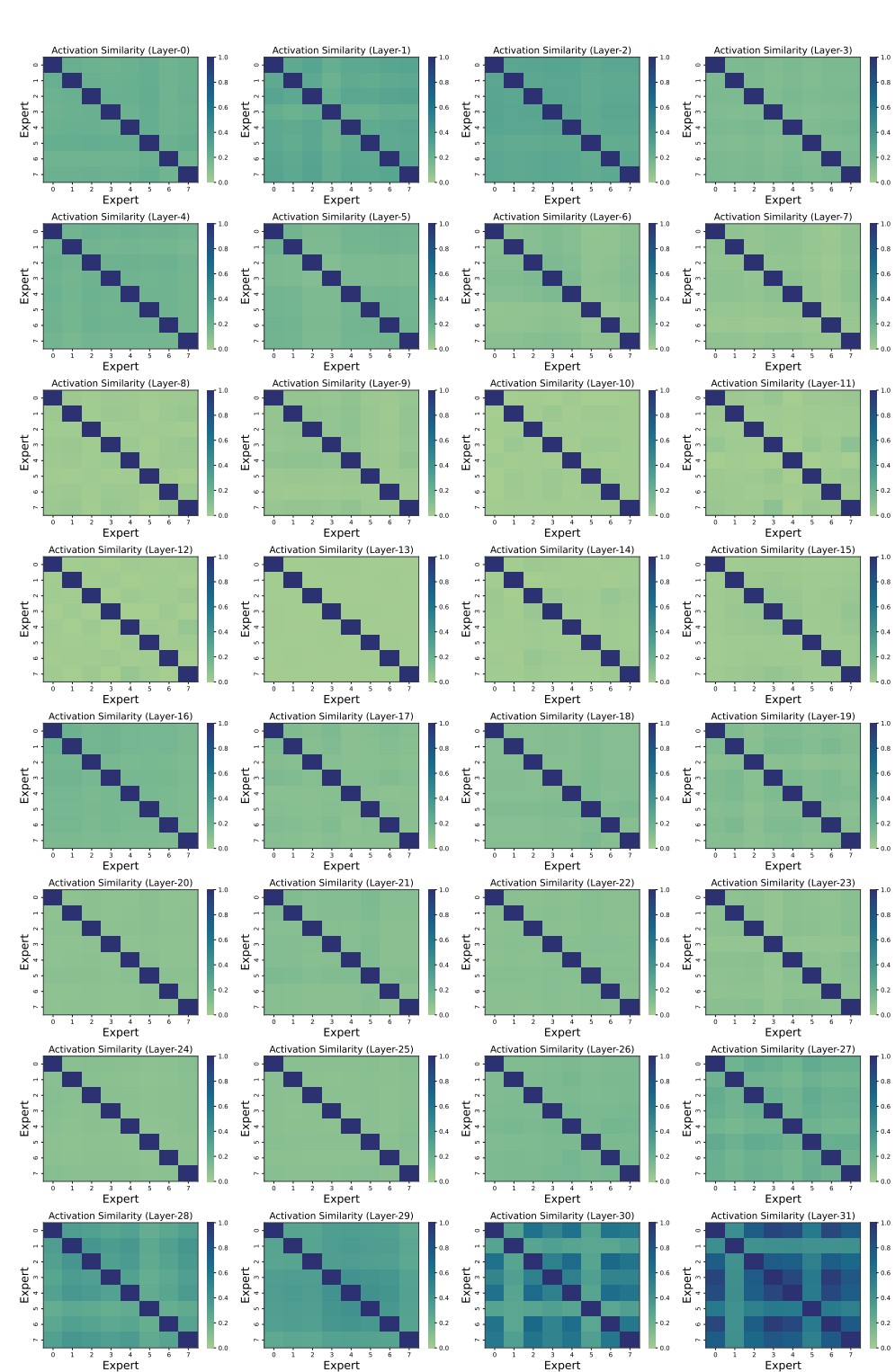

Figure 12: **Experts Activation Similarity (EAS):** Snapshot of Expert-Expert Activation similarity estimated using C4 dataset for Mixtral-8×7B `Base` model. Least dominant expert are identified as expert which have highest similarity with rest of other experts within corresponding layer.

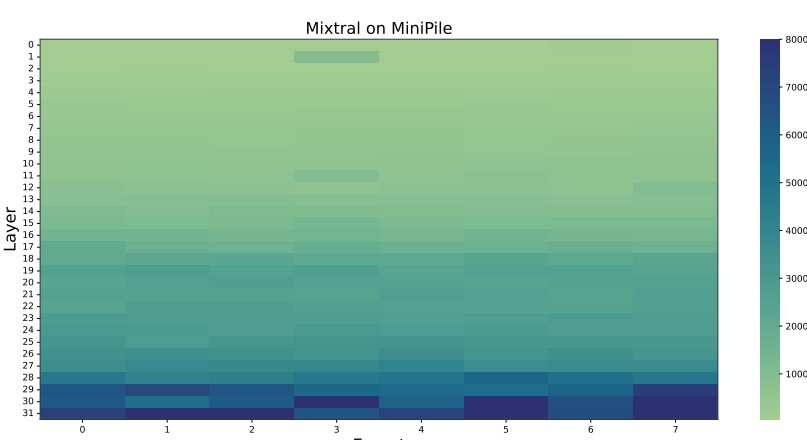

Figure 13: **Experts Activation Entropy (EAE):** Heatmap corresponding to activation entropy estimated for different experts using C4 dataset for Mixtral-8×7B Base model. Interesting, we find that activation entropy gradually increases as we move from intial layers to terminal MoE layers. Experts with minimal activation entropy within a MoE layer are better candidates for dropping. Note that even in some initial layers, it can be observed that some experts carry notable entropy and dropping them lead to significant performance degradation.

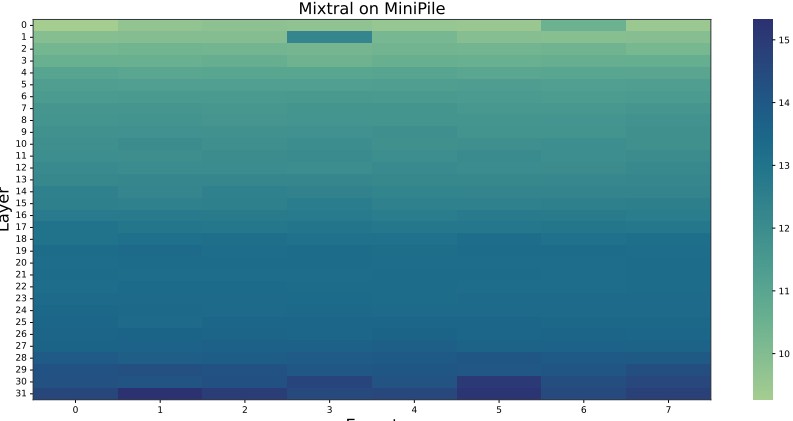

Figure 14: **Experts Gradient Entropy (EGE):** Illustration of the gradient entropy estimated using C4 dataset for Mixtral-8×7B Base model. We found a strong positive co-relation between the experts with high activation entropy and gradient entropy. Similar to activation entropy, we found two experts in Layer 1 and 2 of the checkpoint having significantly high gradient rntropy and dropping them lead to abrupt performance drop.

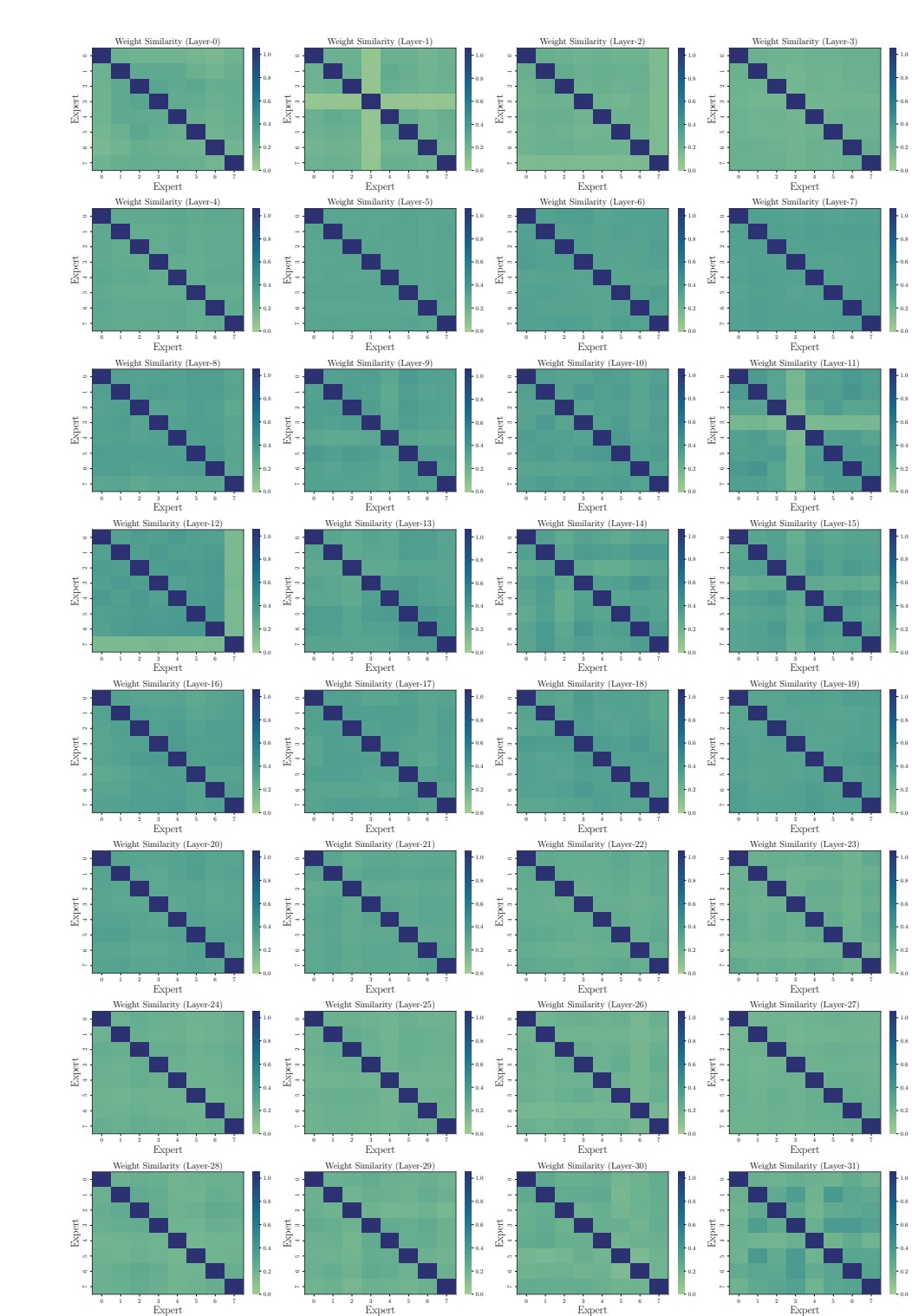

Figure 15: **Experts Weight Similarity (EWS):** Heatmap illustrating the weigh similairty acorss 8 experts corresponding to 32 MoE layers of Mixtral-8×7B Base model. Expert with highest weight similarity across remaining 7 experts becomes the better candidate for expert dropping.

