# OpenReview forum: "Fantastic Experts and How to Find Them: A  Multi-Dimensional Study for Experts-Level Sparsification in Mixture-of-Experts"
_ICLR.cc/2025/Conference — Submitted to ICLR 2025_

### Official Review · Reviewer_kRxK · 2024-11-02

**Soundness:** 2
**Presentation:** 2
**Contribution:** 1
**Rating:** 3
**Confidence:** 5

**Summary:**

This paper presents a comprehensive study of expert-level sparsification in Mixture-of-Experts (MoE) models. The authors propose MC-Suite, a collection of criteria for identifying and pruning less important experts, spanning weight-guided, inference behavior, activation-guided, and gradient-guided perspectives. They also introduce iterative pruning with task-agnostic fine-tuning and investigate the impact of expert pruning on model capabilities. The approach is evaluated on various MoE models including Mixtral-8×7B and demonstrates competitive performance compared to existing pruning methods.

**Strengths:**

1. **Problem Significance:**
  Addresses an important practical challenge of reducing memory requirements in MoE models.
  The multi-dimensional study provides insights into expert importance from different perspectives.

2. **Technical Insights:**
  The observation that instruction-following capabilities are predominantly affected during expert pruning is interesting.
  The empirical findings about the connection between expert characteristics and model performance provide valuable insights.

3. **Experimental Breadth:**
 Comprehensive evaluation across multiple models and tasks.
 Detailed ablation studies examining different components of the proposed approach.

**Weaknesses:**

1. **Methodological Foundation:**
  1.1. Lacks theoretical justification for the proposed criteria in MC-Suite: The activation-based criteria (EAS, EAE) lack clear connection to expert importance; The weight-based metrics (EWS, RWN) need theoretical grounding for why they indicate expert redundancy; No justification for why gradient-based measures reliably indicate expert importance;
  1.2. The iterative pruning and fine-tuning approach appears ad-hoc without strong motivation: No clear explanation why k rounds is better than one-shot pruning; The choice of pruning schedule (amount pruned per round) seems arbitrary;The relationship between pruning rounds and fine-tuning steps needs justification.
   1.3. Many design choices seem empirically driven without proper justification, some of them are listed as follows: Selection of 32 experts per round for pruning; Choice of task-agnostic fine-tuning dataset; Hyperparameter settings for reconstruction compensation.


2. **Technical Presentation:**
  2.1. Missing a dedicated Related Work section in the main body, making it difficult to understand the paper's positioning
  2.2. Introduction is unnecessarily long and unfocused.
  2.3. The methodology section lacks clear organization and proper explanation of key components.
  2.4. Mathematical notations and formulations are not well explained (e.g., equations 3-18).

3. **Experimental Design:**
  3.1. Limited comparison with state-of-the-art baselines, especially on some key metrics like perplexity. The comparison with following ones are necessary: Recent expert pruning methods like SEER-MoE; Standard model compression techniques adapted for MoE; Performance on more established benchmarks.
  3.2. Experiments on larger MoE models (>8x7B parameters) are missing.
  3.3. Implementation details are insufficient for reproducibility, some of them are crucial: Exact hyperparameters for fine-tuning (learning rate schedule, batch size, etc.); Details of the calibration process for different criteria; Computational infrastructure requirements.
  3.4. Ablation studies don't fully justify the necessity of all proposed components, the following ones should be specifically examined: Individual contribution of each criterion in MC-Suite; Impact of different pruning schedules; Effect of fine-tuning duration;

4. **Paper Organization and Clarity:**
  4.1. Figures lack proper resolution and detailed captions (e.g., Figure 1).
  4.2. Inconsistent font sizes and styles in figures (particularly Figure 2).
  4.3. Writing style is verbose with low information density, some parts need more concision: Literature review can be shortened by 40%; Experimental setup description is repetitive;Results discussion contains redundant information.
  4.4. Poor organization makes it difficult to follow the main contributions, some suggestions can be considered: The authors can move technical background to preliminaries, consolidate criteria description into a unified framework, and present results in a more structured manner.

**Questions:**

1. **Methodology:**
  1.1. How do you justify the selection of these specific criteria in MC-Suite? What makes them particularly suitable for expert importance estimation?
 1.2. Why does iterative pruning perform better than one-shot pruning? What's the theoretical basis?
 1.3. Can you provide theoretical bounds on the information loss when removing experts based on different criteria?
 1.4. How does the method handle cases where experts are equally important according to different criteria?


2. **Experimental Design:**
 2.1. Could you provide more detailed comparisons with SOTA methods, especially on perplexity metrics?
 2.2. Have you tested the approach on larger MoE models? What are the scalability limitations?
 2.3. How sensitive is the method to different hyperparameters in the fine-tuning process?
 2.4. What is the relationship between model size and optimal pruning schedule?



3. **Practical Considerations:**
 3.1. What is the computational overhead of the iterative pruning and fine-tuning process?
 3.2. How does the choice of calibration dataset affect the performance of different criteria?
 3.3. Can you provide more details about the implementation of the reconstruction compensation method?

---

### Official Review · Reviewer_CzHe · 2024-11-03

**Soundness:** 2
**Presentation:** 2
**Contribution:** 2
**Rating:** 5
**Confidence:** 4

**Summary:**

This paper presents a comprehensive study of expert-level sparsification in Mixture-of-Experts (MoE) models. The authors propose MC-Suite, a collection of criteria from different perspectives (weight-guided, inference behavior, activation-guided, and gradient-guided) to identify and prune less important experts. They also introduce iterative pruning with task-agnostic fine-tuning and investigate how expert pruning impacts model capabilities. The approach is evaluated on various MoE models including Mixtral-8×7B, demonstrating improvements over existing pruning methods.

**Strengths:**

1. The paper tries to address the practical challenge of MoE memory efficiency through a systematic study of expert importance.
2. The introduced MC-Suite provides a comprehensive collection of criteria from different perspectives.
3. Some empirical findings about expert characteristics and behavior are potentially valuable.
4. The evaluation considers multiple aspects including consistency, accuracy, and efficiency.

**Weaknesses:**

1.  Most criteria in MC-Suite lack originality and proper justification. EWS, EAN, and other metrics are direct adaptations from traditional pruning methods without convincing evidence of their applicability to expert importance estimation. The gradient-     based criteria introduce significant computational overhead without demonstrating clear benefits.
2. The iterative pruning framework has multiple unaddressed issues such as lack of stopping criteria and potential catastrophic forgetting. The task-agnostic fine-tuning could potentially harm expert specialization. The linear combination assumption in reconstruction compensation oversimplifies the complex relationships between experts.
3. The experimental evaluation is incomplete. Critical comparisons with state-of-the-art methods (e.g., ExpertPrune, DynamicMoE) are missing. The paper doesn't examine zero-shot generalization, impact on load balancing, or changes in expert specialization patterns. The largest tested model is only 8×7B parameters. The papers for ExpertPrune and DynamicMoE are listed as follows:  ExpertPrune: Lu, Xudong, et al. "Not All Experts are Equal: Efficient Expert Pruning and Skipping for Mixture-of-Experts Large Language Models." arXiv preprint arXiv:2402.14800 (2024); DynamicMoE: Huang, Quzhe, et al. "Harder Tasks Need More Experts: Dynamic Routing in MoE Models." arXiv preprint arXiv:2403.07652 (2024). There are actually many related works in recent years, which should be compared in the experiments.
4. Implementation details crucial for reproducibility are missing in the main body. The paper doesn't specify key hyperparameters for fine-tuning, details of the calibration process, or computational requirements.
5. The analysis of expert behavior changes during and after pruning is superficial. There's no examination of how pruning affects routing mechanisms, token distribution, or model robustness. A more detailed case study should be provided.
6. The figures have severe presentation issues: Figure 1 has poor resolution and lacks the necessary caption for explaining the figure content.  Figure 2's text uses inconsistent fonts and sizes. Figure 4 contains too much context. Its organization can also be further improved.
7. The writing style in this paper is not easy to read and understand. Some proposed concepts, like 'MOE Lottery Subnetworks', have vague meanings. The core technique motivation and details, however, are not clarified in detail. The organization of this paper, especially Section 2 about methodology, is over-scattered, and it is hard to capture the logical structure and the most significant contribution. The presentation lacks a clear separation between methodology and experimental validation, making it challenging to distinguish the proposed method from its empirical evaluation. A more structured organization with clearly demarcated methodology and experiment sections would significantly improve readability. The proposed equations in this paper lack corresponding explanations or clarifications.
8. Several language and formatting issues throughout the paper:  Typo: "idetified" instead of "identified" in footnote 3; Misplaced comma in "Our experiments found that a non-uniform dropping of experts per layer by estimating c globally creates bottleneck layers, with some layers having significantly high sparsity while some remain unpruned, leading to diminished finetuning benefits and sharding simplicity."; Missing commas or periods after equations 1-18;  Missing articles: "by gating function" should be "by the gating function" (line 232)

**Questions:**

1. How do you justify using weight similarity when experts are supposed to learn different patterns?
2. What guarantees the preservation of expert specialization after linear reconstruction?
3. How does iterative pruning affect the router's ability to dispatch tokens effectively?
4. Why use task-agnostic fine-tuning when experts are task-specialized?
5. How do you determine optimal pruning schedules?
6. What measures prevent catastrophic forgetting during fine-tuning?
7. How does computational cost scale with model size?
8. What is the memory overhead during pruning/fine-tuning?
9. How to handle cases where criteria give conflicting importance scores?

---

### Official Review · Reviewer_fTQR · 2024-11-11

**Soundness:** 3
**Presentation:** 2
**Contribution:** 3
**Rating:** 5
**Confidence:** 4

**Summary:**

This paper proposes the MoE Experts Compression Suite (MC-Suite), which includes a collection of previously proposed and some novel expert importance estimation criteria for expert-level pruning in SMoE models. The authors provide an extensive empirical study of different pruning criteria on the Mixtral 8-7B Base model. Additionally, they propose MoE lottery pruning and demonstrate the benefits of iterative pruning-retraining in maintaining or improving downstream task performance compared to one-shot expert pruning techniques. Finally, through their empirical studies, the authors argue that expert pruning predominantly harms the instruction-following capabilities of SMoEs, but this loss can be potentially restored with external instruction-following guidance.

**Strengths:**

- The paper proposes a variety of new expert pruning criteria and offers a comprehensive study on the empirical performance of a wide array of pruning criteria applied to the Mixtral 8-7B Base model.
- The proposed MoE Lottery Pruning method is straightforward, and the task-agnostic fine-tuning component shows empirical effectiveness.

**Weaknesses:**

I am open to discussion and willing to reconsider my score if my major concerns are resolved.

- Some experimental details are missing or hard to find. Please refer to the questions for more details.
- The empirical studies, particularly those concerning the identification of the most effective expert-pruning criteria (e.g., Table 2), were conducted on a single pre-trained MoE model, specifically the Mistral 8x7B Base. This limitation could potentially undermine the validity of the empirical findings in Section 3.1, as it remains unclear whether these conclusions are generally applicable.
- The expert pruning criteria appear suitable only for considering expert pruning within each layer independently. As stated in line 375, non-uniform dropping of experts leads to poorer performance.

**Questions:**

1. **Experiment details.** I couldn't find some important details regarding the implementation of the proposed "MoE Lottery Pruning." Specifically:

   - Task-agnostic fine-tuning: What dataset was used for this fine-tuning procedure? Based on the appendix, it seems like it's the validation set of C4. Could the authors please clarify?

   - Table 3 and Figure 6: Which model was used in these experiments, the base model or the instruct model?

   - Figure 6, the third row, where the models are supervised fine-tuned: What instruction-tuning datasets were used? Are they the ones listed in Table 6 of the appendix? If so, for each dataset, are the models fine-tuned on the respective training split then?
   -  In line 412, "Interestingly, while comparing...." Could the author please clarify the source of this observation? It seems that this comparison refers to Table 1 (instruct model) and Table 2 (base model), but the caption of Table 2 does not explicitly state which model is used in the experiments.

2. **Is task-agnostic fine-tuning really effective for an pruned *Instruct* model?**

    - Why was perplexity compared in Table 1 for an *instruction-tuned* model on a pre-training dataset like C4? Is it a fair comparison, considering that MoE lottery pruning may have been task-agnostically fine-tuned on a validation set of C4 (if this is the case)?
    - Were there any experiments in the paper that demonstrate the downstream performance of an instruct model benefiting from the addition of the proposed task-agnostic fine-tuning on C4? For instance, something similar to what is shown in Table 3, or with in-context learning examples instead of zero-shot?

3. **Calibration dataset for pruning criteria.** In line 216, the authors mentioned, "we experimentally found that expert usage frequency is not strongly tied to the choice of calibration dataset." This is an interesting observation, but is it supported by empirical evidence elsewhere in the paper? Furthermore, when pruning criteria require a calibration dataset, such as for gradients and activations based criteria, does this insensitivity to the calibration dataset, which was specifically noted for the EUF criteria, still apply in general?

4. **Technical novelty in MoE Lottery Pruning.** Iterative pruning-retraining itself is not a new technique, as it has been used in traditional weight pruning research, e.g.,(the lottery ticket hypothesis). Regarding the retraining aspect, it appears that the use of retraining (or task-agnostic fine-tuning in this work) is a direct application of the retraining technique from the lottery ticket paper. The difference lies in addressing suboptimality in the router networks and the pruned experts, rather than in the pruned weights. However, I admit the MoE lottery pruning approach seems novel in the context of expert-level pruning. Could the authors potentially discuss whether this is a straightforward application of a previously proposed technique? It would be very beneficial if the authors could highlight the major differences from some previous iterative pruning-retraining techniques and perhaps some of the unique challenges faced when applying this technique in the context of expert-level pruning.

---

### Meta-Review · Area_Chair_HwWn · 2024-12-22

**Metareview:**

This paper introduces the MoE Experts Compression Suite (MC-Suite), which combines previously proposed and novel expert importance estimation criteria for expert-level pruning in SMoE models. These criteria include weight-guided, inference-behavior, activation-guided, and gradient-guided approaches. The method is evaluated on various MoE models, including Mixtral-8×7B, with iterative pruning and task-agnostic fine-tuning to explore the impact of expert pruning on model capabilities.

The strengths recognized by all reviewers include (1) empirical findings on expert characteristics and behaviors, and (2) a comprehensive collection of pruning criteria from multiple perspectives.

However, the current work appears unprepared for publication, as evidenced by unanimous rejection from all reviewers. Major weaknesses include (1) limited technical novelty, with most criteria in MC-Suite and MoE Lottery Pruning being non-novel; (2) incomplete experimental evaluations, such as missing critical comparisons with state-of-the-art methods, insufficient exploration of generalization to larger MoE models, and inadequate details for reproducibility; and (3) poor presentation quality.

**Additional Comments On Reviewer Discussion:**

The authors did not provide any response during rebuttal.

---

### Decision · Program_Chairs · 2025-01-22

Reject